# Selective CDK9 Inhibition by Natural Compound Toyocamycin in Cancer Cells

**DOI:** 10.3390/cancers14143340

**Published:** 2022-07-08

**Authors:** Somnath Pandey, Rahinatou Djibo, Anaïs Darracq, Gennaro Calendo, Hanghang Zhang, Ryan A. Henry, Andrew J. Andrews, Stephen B. Baylin, Jozef Madzo, Rafael Najmanovich, Jean-Pierre J. Issa, Noël J.-M. Raynal

**Affiliations:** 1Fels Institute for Cancer Research and Molecular Biology, Temple University School of Medicine, Philadelphia, PA 19140, USA; sxp3248@med.miami.edu (S.P.); hanghangzhang@temple.edu (H.Z.); jmadzo@coriell.org (J.M.); jpissa@coriell.org (J.-P.J.I.); 2Département de Pharmacologie et Physiologie, Université de Montréal, Montréal, QC H3T 1J4, Canada; rahinatou.djibo@umontreal.ca (R.D.); anais.darracq@umontreal.ca (A.D.); rafael.najmanovich@umontreal.ca (R.N.); 3Sainte-Justine University Hospital Research Center, Montréal, QC H3T 1C5, Canada; 4Coriell Institute for Medical Research, Camden, NJ 08103, USA; gcalendo@coriell.org; 5Department of Chemistry and Biochemistry, Wilkes University, Wilkes-Barre, PA 18766, USA; ryan.henry@wilkes.edu; 6Department of Pharmaceutical Sciences, Wilkes University, Wilkes-Barre, PA 18766, USA; 7Department of Cancer Biology, Fox Chase Cancer Center, Philadelphia, PA 19111, USA; andrew.andrews@fccc.edu; 8The Sidney Kimmel Comprehensive Cancer Center at Johns Hopkins, Baltimore, MD 21231, USA; sbaylin@jhmi.edu

**Keywords:** CDKs, CDK9 inhibitor, toyocamycin, drug screening, epigenetics, cancer, RNA Pol II phosphorylation, molecular docking

## Abstract

**Simple Summary:**

By combining drug screens, transcriptomic studies, and in vitro assays, our study identified the natural product toyocamycin as a potent and selective CDK9 inhibitor. Thus, toyocamycin can be used as a new small molecule to modulate CDK9 activity in preclinical research. Through docking simulations, we identified its specific binding sites, which could spark some interest to design novel small molecule CDK9 inhibitors.

**Abstract:**

Aberrant transcription in cancer cells involves the silencing of tumor suppressor genes (TSGs) and activation of oncogenes. Transcriptomic changes are associated with epigenomic alterations such as DNA-hypermethylation, histone deacetylation, and chromatin condensation in promoter regions of silenced TSGs. To discover novel drugs that trigger TSG reactivation in cancer cells, we used a GFP-reporter system whose expression is silenced by promoter DNA hypermethylation and histone deacetylation. After screening a natural product drug library, we identified that toyocamycin, an adenosine-analog, induces potent GFP reactivation and loss of clonogenicity in human colon cancer cells. Connectivity-mapping analysis revealed that toyocamycin produces a pharmacological signature mimicking cyclin-dependent kinase (CDK) inhibitors. RNA-sequencing revealed that the toyocamycin transcriptomic signature resembles that of a specific CDK9 inhibitor (HH1). Specific inhibition of RNA Pol II phosphorylation level and kinase assays confirmed that toyocamycin specifically inhibits CDK9 (IC_50_ = 79 nM) with a greater efficacy than other CDKs (IC_50_ values between 0.67 and 15 µM). Molecular docking showed that toyocamycin efficiently binds the CDK9 catalytic site in a conformation that differs from other CDKs, explained by the binding contribution of specific amino acids within the catalytic pocket and protein backbone. Altogether, we demonstrated that toyocamycin exhibits specific CDK9 inhibition in cancer cells, highlighting its potential for cancer chemotherapy.

## 1. Introduction

Epigenetic mechanisms play important roles in cancer cells in creating and maintaining aberrant gene expression profiles, such as the silencing of tumor suppressor genes (TSGs) or the activation of oncogenes [1,2]. Epigenetic mechanisms include processes such as DNA methylation, histone modifications, and chromatin remodeling that govern heritable gene expression patterns without any changes in the underlying DNA sequence [3,4]. Acquisition of DNA methylation and/or chromatin repressive marks, at the promoter of TSGs, results in gene silencing via recruitment of the methyl-binding domain (MBD) containing proteins such as MeCP2 or by transcription factor binding inhibition [5,6]. Then, repressor complexes, including histone deacetylases (HDACs), causing a loss of histone acetylation, and chromatin remodelers, inducing heterochromatin formation, are recruited to induce long-term gene silencing [6,7].

Given the reversible nature of these epigenetic modifications, epigenetic therapy of cancer aims to reverse such alterations and induce TSG reactivation, resulting in cancer cell differentiation and apoptosis and recognition by the immune cells [2,3,6]. Epigenetic drugs are approved for the treatment of hematological malignancies and occasional proof-of-principle responses have been observed in solid tumors [6,8]. Treatment options are, however, limited to a small number of clinically approved epigenetic drugs with four HDAC inhibitors (HDACis; vorinostat, romidepsin, belinostat, and panobinastat), two DNA methylation inhibitors (DNMTis; azacitidine and decitabine), and more recently, EZH2 (tazemetostat) and IDH1/2 inhibitors (enasidenib and ivosidenib) [9,10,11].

Drug discovery initiatives are underway in rare and specific cancer types with well-defined mutations in epigenetic effectors. However, these efforts may take years before approval and may have limited effects outside of a restricted patient population [12]. There is an urgent need to discover novel epigenetic drugs to tackle the cancer epigenome with different mechanisms of action [9]. Phenotypic or target-based assays are commonly used in drug discovery. As a platform for epigenetic drug screening, we developed a phenotypic assay using the established YB5 cell-based system, which is derived from SW48 human colon cancer cell line [13,14]. The YB5 system contains a single insertion of a transgene comprising the green fluorescent protein (*GFP*) gene driven by the cytomegalovirus (*CMV*) promoter. In >99.9% of YB5 cells, GFP expression remains silent as a result of the *CMV* promoter being subjected to DNA hypermethylation and embedded in a closed chromatin structure marked by repressive histone 3 lysine 27 (H3K27) hypermethylation and H3K9 deacetylation [14]. As such, in YB5 cells, *GFP* phenocopies endogenously silenced TSGs by epigenetic mechanisms and can be re-expressed following treatment with a DNMTi and/or an HDACi [13,14]. Therefore, the YB5 system is a useful cell-based model to discover novel epigenetic drug targets that reverse tumor suppressor gene silencing [15], but also to identify new pathways of gene reactivation [15,16,17]. In our pursuit to discover novel epigenetic drugs that can be rapidly tested in clinical trials, we have been screening multiple drug libraries (FDA-approved and anticancer libraries) and optimizing positive hits [15,16,17,18,19]. Natural compounds are key tools in preclinical cancer research and some of them have been approved for cancer treatment (e.g., taxol, romidepsin). There is a benefit to screening natural product libraries since approximately 40% of the natural product chemical scaffolds are absent in the contemporary medicinal chemistry repertoire [20]. An earlier report, using the YB5 system, highlighted that a natural product that accounts for about 5% of royal jelly, namely 10-hydroxy-2-decenoic acid, triggers TSG reactivation through histone deacetylase inhibitory activity [21]. Thus, it is of particular interest to screen natural compounds in efforts to discover novel epigenetic drugs.

In the current study, we performed an unbiased epigenetic drug screening for GFP reactivation using YB5 cells, with a natural compound drug library. We identified a natural compound, referred to as toyocamycin (7-Deaza-7-cyanoadenosine), previously reported for its inhibitory role in endoplasmic reticulum stress-induced XBP1 mRNA splicing [22]. We showed that toyocamycin causes stronger gene reactivation than approved epigenetic drugs and showed similar patterns of gene regulation to a previously reported cyclin-dependent kinase 9 (CDK9) inhibitor HH1 [15]. We tested its inhibitory effects, binding site, and specificity to CDK9, as compared to other CDKs.

## 2. Materials and Methods

### 2.1. GFP Cell-Based System, Natural Drug Library, and Drugs

YB5 cells are derived from SW48 cells (human colon cancer cell line), and contain a single insertion of a DNA hypermethylated cytomegalovirus (*CMV*) promoter driving the green fluorescent protein (*GFP*) gene, as previously described [14]. YB5 cells are cultured in L-15 medium supplemented with 10% fetal bovine serum and grown in log-phase in 1% CO_2_ atmosphere. YB5 cells were authenticated by DNA fingerprinting prior to drug screening and validation experiments. *GFP* expression in YB5 cells is silenced in >99.9% of the cells due to the *CMV* promoter DNA hypermethylation and repressive chromatin marks around the transcriptional start site [13,14,15,16,17]. The human HCT116 colon cancer-derived GFP-reporter system, comprising a *GFP* gene under the control of an endogenously hypermethylated *SFRP1* gene promoter, was kindly provided by Dr. Stephen Baylin (Johns Hopkins University) [23]. These cells are cultured in McCoy’s 5A medium supplemented with 10% fetal bovine serum and grown in log-phase in 5% CO_2_ atmosphere at 37 °C. The 451-Lu-BR melanoma cells were a kind gift from Dr. Raza Zaidi (Fels Institute for Cancer Research and Molecular Biology, Temple University School of Medicine, Philadelphia). These cells are derived from parental 451-LU cells that have acquired resistance to BRAF inhibitors. The 451-Lu-BR cells were cultured at 37 °C in Dulbecco’s Modified Eagle Medium (DMEM) supplemented with 10% fetal bovine serum, L-alanyl-L-glutamine (2 mM), and gentamycin (50 µg/mL) in 5% CO_2_ at 37 °C. The natural drug library (natural product set II) containing 120 compounds was obtained from the National Cancer Institute Developmental Therapeutics Program. Natural compounds were dissolved in DMSO (10 mM) in 96-well plates that were kept at −80°C until use. Toyocamycin, analogues of toyocamycin (sangivamycin, tubercidin, 3-deaza-adenosine, DZNep), CDK inhibitors, and epigenetic drugs (decitabine, SAHA, trichostatin A, depsipeptide) were purchased from Sigma-Aldrich (St. Louis, MO, USA,) for validation studies, dissolved in DMSO, and kept at −80 °C until use.

### 2.2. Drug Screening Conditions

YB5 cells were seeded at 30,000 cell/mL in 96-well plates (6000 cells/well) and treated 3 days after seeding. Cells were exposed to 120 natural compounds in 5 different experimental conditions: (1) 72 h exposure at 10 µM followed by 24 h recovery period prior to analysis; (2) 24 h treatment at 50 µM prior to analysis; (3) 72 h exposure at 10 µM of the compound library followed by 24 h treatment with HDAC inhibitor trichostatin A (200 nM) prior to analysis; (4) 72 h exposure at 10 µM in simultaneous combination with decitabine (50 nM) followed by 24 h recovery period prior to analysis; and (5) 72 h treatment with decitabine (50 nM) followed by 24 h exposure with the natural compound library (50 µM) prior to analysis. The scheme of different dose schedules is shown Appendix A. During screening, drugs and cell culture media were replaced every day. To ensure the quality and the reproducibility of the screen, each 96-well plate was designed to include 80 natural compounds, 4 untreated wells, 4 wells treated with decitabine (50 nM, 72 h), 4 wells treated with trichostatin A (200 nM, 24 h), and their combination with decitabine (50 nM, 72 h) followed by trichostatin A treatment (200 nM, 24 h) prior to flow cytometry analysis. The percentage of GFP-expressing cells in control wells (untreated or treated with known epigenetic drugs) was measured and was reproducible between different plates.

### 2.3. Flow Cytometry for GFP Expression and Data Analysis

After treatment, YB5 cells were trypsinized for 15 min and resuspended in L-15 media containing propidium iodide (PI) to stain dead cells. GFP expression and PI staining were quantified by a BD LSR II flow cytometer (BD Biosciences, San Jose, CA, USA). A maximum of 10,000 cells was analyzed per well. Validations were performed using a Millipore Guava flow cytometer (EMD, Millipore, Burlington, MA, USA). As described previously [16], autofluorescent compounds were removed from the analysis (showing a high intensity signal in the double positive quadrant for both GFP and PI). Based on a previous study [16], positive hits in monotherapy were selected as those non-autofluorescent compounds producing more than 2.2% of GFP^+^ cells. In the combinatorial screen, GFP expression produced by the natural compounds was compared to the effects of decitabine alone (producing about 15% of GFP^+^ cells at 50 nM, 72 h) and trichostatin A alone (producing about 6% of GFP^+^ cells at 200 nM, 24 h). For validation studies, dose-response experiments were performed using PI to stain for dead cells in YB5 and HCT116-GFP models.

### 2.4. DNA Methylation Analysis by Pyrosequencing

We analyzed DNA methylation levels on the *CMV* promoter region and *LINE-1* repeated elements (as a marker of global DNA methylation level). After DNA extraction and bisulfite conversion, PCR amplification of regions of interest and pyrosequencing were achieved as previously described [13,14].

### 2.5. Histone Acetylation Analysis by Mass Spectrometry

As described previously [16,24], histone proteins were acid-extracted (0.2 N HCl) for 16 h. Histone acetylation levels were quantified on histone 3 and histone 4 by a Thermo TSQ Quantum Access triple quadrupole (QqQ) mass spectrometer (Thermofisher Scientific, Waltham, MA, USA). 

### 2.6. Anchorage-Dependent Clonogenic Assays

Two hundred YB5 cells were seeded per well in 6-well plates. On the following day, YB5 cells were treated with toyocamycin for 24 h or 72 h. Decitabine (10–50 nM for 72 h) was used as a positive control for the loss of clonogenic potential. Drugs and media were replaced each day. After treatment, media was removed. YB5 cells were grown in drug-free media for 2 weeks to allow visible colony formation prior to colony staining (0.5% methylene blue in 50% methanol). YB5 colonies were counted manually.

### 2.7. RNA Sequencing and Transcriptomic Analysis

Sequenced reads were aligned to the GRCh38.100 reference genome using STAR version 2.7.5a [25] and gene-level counts were generated by setting the “quantMode GeneCounts” flag. Differential expression analysis was performed using EdgeR version 3.32.1. [26]. The “filterByExpr” function in EdgeR was then used to determine genes with sufficiently large counts to be included in downstream differential expression testing. The filtered count matrix was then normalized using TMM normalization and differential expression testing was performed using the “glmTreat” function with the minimal required fold-change threshold set to “log2(1)” for significant differential gene expression (FDR < 0.1).

### 2.8. Cell Viability, Proliferation Assays, and Cell Cycle Analysis

Cell viability was measured in YB5 and HCT166 cells using Viacount (Luminex, Austin, TX, USA, 4000-0040) on a Guava flow cytometer (Millipore, Burlington, MA, USA). Colon cancer cells were grown in 96-well plates in the same conditions as in the drug screen and treated with toyocamycin and known CDK9 inhibitor BAY1251152 (Selleckchem, Houston, TX, USA). Cell culture media of each well was collected to harvest floating dead cells and kept aside for later. Cells were trypsinized (0.25% trypsin, Gibco, Waltham, MA, USA) for 5 min at 37 °C. Cell culture media of each well was added back to their corresponding wells to stop trypsinization and mixed thoroughly to obtain a single cell suspension. Viacount solution (Luminex, Austin, TX, USA, 4000-0040, 25 μL) was added to each well, and incubated (5 min at room temperature, in the dark) prior to cell viability analysis on the Guava flow cytometer.

Proliferation assays were performed for 451-Lu-BR human melanoma cells in response to toyocamycin exposure. First, 20,000 cells were seeded per well in 24-well plates. Cells were treated with either DMSO (<0.01% final concentration) or various toyocamycin concentrations. Cell proliferation was assessed by cell counting using a hemocytometer (*n* = 3) after 24, 48, and 72 h of treatment. The data were plotted as the total number of cells in response to toyocamycin for each treatment duration relative to DMSO treated cells.

For cell cycle analysis, cells were cultured in T25 flasks, trypsinized, and washed with PBS. Then, 1 × 10^6^ cells were fixed in 100% cold ethanol overnight at −20 °C. Cells were then washed with PBS and treated with 100 μg/mL RNase A (Invitrogen, Waltham, MA USA) for 20 min, at 37 °C. DNA was stained with 5 μg/mL propidium iodide (Sigma, St. Louis, MO, USA). The cell cycle was analyzed by flow cytometry using a Guava^®^ easyCyte HT System (Millipore) with GuavaSoft 3.2 software.

### 2.9. Cyclin-Dependent Kinase Inhibition Assays

Enzymatic activities of recombinant human CDK9/Cyclin T1, CDK2/Cyclin 2A, CDK4/Cyclin D3, CDK6/Cyclin D3, and CDK7/Cyclin H/MAT1 were tested against toyocamycin and known CDK inhibitors (staurosporine, dinaciclib, palbociclib). The in vitro enzymatic assays were conducted at BPS Bioscience. CDK9/Cyclin T1, CDK2/Cyclin 2A, CDK4/Cyclin D3, and CDK6/Cyclin D3 enzymatic activities were tested with an ATP Kinase-Glo Plus Luminescence kinase assay kit (Promega, V6073, Madison, WI, USA), while the CDK7/Cyclin H/MAT1 activities were tested using ADP-Glo Kinase (Promega, V6930). These assays measure kinase activity by quantifying the amount of ATP or ADP remaining in solution following a kinase reaction. The luminescent signal from the assay is correlated to the amount of ATP or ADP present and is inversely correlated to the kinase activity. The compounds were diluted to 10% DMSO and were added to the reaction so that the final concentration of DMSO was 1% in all reactions. All enzymatic reactions were conducted at 30 °C for 60 min. The reaction mixture contained 40 mM Tris, pH 7.4, 10 mM MgCl2, 0.1 mg/mL BSA, 1 mM DTT, 10 µM ATP, kinase substrate, and the enzyme. After the enzymatic reaction, the luminescence signal was measured using a BioTek Synergy 2 microplate reader (Agilent, Santa Clara, CA, USA). Kinase activity assays were performed in duplicate at each concentration and the percent of kinase activity was calculated as a ratio of luminescence intensities in the absence or in presence of the compounds. The values of percent activity versus a series of compound concentrations were plotted using non-linear regression analysis of the sigmoidal dose-response curve. Data were repeated three times for CDK9 and two times for the rest of the CDKs. IC_50_ values were determined by the concentration causing a half-maximal percent activity (BPS Bioscience, San Diego, CA, USA).

### 2.10. Western Blot and Plasmid Constructs

YB5 cells were harvested in 50 mM Tris-HCl, 1% NP-40, 1 mM EDTA, 1 mM EGTA, 0.25% sodium deoxycholate, 150 mM NaCl, 50 mM sodium fluoride, and 1X Complete protease cocktail (Roche, Basel, Switzerland) with 1 mM PMSF. The total protein concentration was measured in all samples by standard BCA protocol (Pierce-23225, Thermo Scientific, Waltham, MA, USA) to ensure that an equal amount of protein was added to each sample. Protein extracts were run on 12% SDS–PAGE precast gels (Bio-Rad, Hercules, CA, USA, 345-0118). Proteins were transferred to polyvinylidene difluoride (PVDF) membranes in 10 mmol/L CAPS (pH-11) containing 10% methanol and detected by using specific primary antibodies, horseradish peroxidase–conjugated secondary antibodies (GE Healthcare, Chicago, IL, USA), and an enhanced chemiluminescence reagent (Pierce). PVDF membranes were incubated with specific primary antibodies (actin—A5441, Sigma, St. Louis, MO, USA; HA-tagged—H9608, Sigma; FLAG-tagged—F3165, Sigma). FLAG- and HA-tagged CDK9 plasmids were a generous gift from Dr Bassel E. Sawaya (Temple University). For CDK9 target modulation, HCT116 colon cancer cells were treated with toyocamycin or known CDK9 inhibitor BAY1251152 (Selleckchem, Houston, TX, USA). Whole cell extracts were obtained by lysing cells with 25 mM Tris (pH 7.3) and 1% sodium dodecyl sulfate (SDS). After sonication, total protein concentrations were determined using Bradford protein assay. The following primary antibodies were used to investigate the effect of CDK9 inhibition: Pol II against the N-term region (sc-55492, Santa Cruz, Dallas, TX, USA), Pol II against the C-term region (39097, Active motif, Carlsbad, CA, USA), Pol II phospho CTD ser2 (14-9802-82, Invitrogen, Waltham, MA, USA), Rb (ab181616, Abcam, Cambridge, UK), Rb phospho T826 (ab133446, Abcam, Cambridge, UK), CDK9 (ab39098, Abcam, Cambridge, UK), and actin (ab1801, Abcam, Cambridge, UK). Images were acquired with an Image Quant LAS4000 device from GE Healthcare Life Sciences. Band intensity quantifications were calculated using ImageJ software, edition 1.53 [27].

### 2.11. Molecular Docking Simulations

The crystal structures of CDK isoforms 2, 6, 7, and 9 were obtained from RCSB Protein Data Bank (PDB) entries: 1GZ8, 5L2T, 1UA2, and 4BCG, respectively. The missing side chains and loop structures of the proteins were modeled using the model/refine loops module in UCSF Chimera. The known 3D structures for CDK4 are in an inactive state [28]. To obtain structural information of active CDK4, homology model structures based on experimentally determined structures of CDK2 and/or CDK6 are often used for computational studies. We used a hybrid model of CDK4, based on the experimentally solved structure of CDK4 (2W96) and CDK2. The 3D conformation of toyocamycin was obtained from the PubChem database. The FlexAID 1.76 software was used to perform docking simulations of toyocamycin on each CDK (29). FlexAID performs flexible docking, which accounts for protein side chain and ligand flexibility using a genetic algorithm as the search procedure. Bound conformations are scored by minimizing the complementarity function (CF) that quantifies protein–ligand interaction-based atomic surface areas in contact using pairwise pseudo-interaction energy terms between 40 atom types and solvent. CF values are in arbitrary units (AU). We evaluated the different docking poses of toyocamycin between different CDKs and the crystal pose of Rio1 kinase by using root mean square deviation (RMSD), calculated with a module included in the FlexAID software [29]. We also calculated the distance between the crystal pose of toyocamycin in Rio1 kinase and the best docking position (best CF value) of toyocamycin in a given CDK, which were expressed as the Best_CF_RMSD in Appendix A. This metric was calculated using the FlexAID software [29].

### 2.12. Statistics

Statistical analysis and graphical representations were performed using the GraphPad Prism 6 software. Student’s *t*-test was used to calculate significance between two groups and the one-way ANOVA test was used to calculate differences for more than two groups.

## 3. Results

### 3.1. Epigenetic Drug Screening with the Natural Product Library

The YB5 cell-based phenotypic assay system (Figure 1A) was utilized to screen a library of 120 natural compounds using five dose schedules (Appendix A). The first schedule involved treating YB5 cells with natural compounds at 10 µM for 72 h followed by a 24 h recovery period without drug prior to analysis. This schedule was crafted to discover epigenetic drugs that require multiple cell divisions to induce robust gene reactivation, as observed with DNMTis [14]. The second schedule comprised treatment of 24 h at 50 µM prior to analysis. This schedule was designed to discover HDACi-like compounds that result in rapid gene reactivation without changes in DNA methylation (Figure 1A) [13,16]. The third schedule comprised 10 µM of compound library treatment for 72 h followed by 24 h treatment with an HDACi, trichostatin A (200 nM), prior to analysis in an attempt to discover drugs which show synergy with HDACis. The fourth schedule involved 10 µM of natural compound library exposure with concurrent treatment with a DNMTi, 5-aza-2′-deoxycytidine (DAC), at 50 nM for 72 h, followed by 24 h of recovery time before analysis. This schedule was selected to identify epigenetic drugs which show synergy with DAC when treated simultaneously [30]. Lastly, the fifth schedule comprised a DAC (50 nM, 72 h) pre-treatment followed by 50 µM of natural compound library prior to analysis. This schedule was created to identify epigenetic drugs, which act once the DNA methylation has been removed from the *GFP* promoter sequence (Figure 1A).

After treatment, YB5 cells were trypsinized and the percentage of GFP-expressing cells was quantified by flow cytometry. Screening results with natural compounds alone and in combination identified toyocamycin (7-Deaza-7-cyanoadenosine) as the only compound producing GFP reactivation in at least three dose schedules out of five (schedules 1, 2, 3) and the best hit in schedule 1 and 3 (Figure 1B and Appendix A-D). Toyocamycin was validated in independent experiments and potently induced GFP fluorescence (Figure 1C). Dose-response curves showed that toyocamycin produced time- and dose-dependent GFP expression. More than 60% of YB5 cells expressed GFP after 5 µM toyocamycin treatment for 72 h (Figure 1C,D). Notably, toyocamycin alone produced more GFP expression than decitabine or SAHA alone (at equivalent doses, Figure 1D). Toyocamycin was also tested in the HCT116 *SFRP1-GFP* model for endogenous TSG reactivation [23]. As in the YB5 cells, toyocamycin induced dose-dependent GFP expression in HCT116 SFRP1-GFP colon cancer cells, further confirming its epigenetic activity (Figure 1E).

To explore the structure–activity relationship (SAR) of toyocamycin-mediated induction of GFP, we tested several adenosine analogues including 3-deaza-adenosine, tubercidin, DZNep, and sangivamycin in the YB5 system. The latter compound is a close structural analogue of toyocamycin [31]. Among those four adenosine analogues, only sangivamycin induced GFP expression in YB5 cells, but to a lesser extent than toyocamycin (Appendix A). Altogether, our epigenetic drug screening identified that toyocamycin triggers epigenetically silenced reactivation in cancer cells. Toyocamycin is an antibiotic isolated from Streptomyces species with anticancer effects and a known inhibitor of ribosomal RNA synthesis [31,32]. However, the epigenetic activity of toyocamycin merits further investigation.

### 3.2. Toyocamycin Induced GFP Expression without Global Changes in DNA Methylation or Histone Acetylation Levels

Drug screen showed that toyocamycin enhanced GFP expression in sequential combination with decitabine (schedule 5, Appendix A), which pointed towards a mechanism targeting chromatin. We confirmed this result in separate experiments with different doses of toyocamycin. The combination of toyocamycin (0.5–50 µM) with decitabine produced GFP expression in up to 40% of YB5 cells (0.5–5 µM, Appendix A).

As toyocamycin alone and in combination with decitabine triggered GFP expression, we first assessed whether it could inhibit DNA methylation levels on the *CMV* promoter in YB5 cells. Using bisulfite pyrosequencing, we found no changes in DNA methylation after toyocamycin treatment (0.05–50 µM, 72 h) at doses producing GFP reactivation (Appendix A). Further confirming this result, *LINE-1* promoter DNA methylation, a marker for global DNA methylation, was not modified by toyocamycin in the same experimental conditions (Appendix A). Since toyocamycin also enhanced GFP expression in sequential combination with decitabine as typically observed with a combination involving HDAC inhibitors [13], we measured histone 3 and 4 acetylation levels by mass spectrometry on eleven lysine residues (Appendix A). Toyocamycin treatments did not modify lysine acetylation levels, suggesting an alternative mechanism of action as compared to currently approved epigenetic drugs. Then, we compared its anticancer effects to an approved DNA demethylating drug (decitabine, DAC) using clonogenic assays. Interestingly, low dose toyocamycin (10 nM, 24 h treatment) completely abrogated colony formation in YB5 cells. At an equivalent dose schedule (10–50 nM, 72 h treatment), toyocamycin fully abolished colony formation while cancer cells still produced colonies after decitabine treatment (Appendix A).

### 3.3. Regulation of Transcription by RNA Polymerase II Is Altered by Toyocamycin Treatment

To decipher the mechanism of toyocamycin associated with gene expression modulation, we used RNA-sequencing on untreated and treated YB5 cells (Figure 2A). We performed a time-course experiment with toyocamycin (250 nM) from 2, 10, 24, 48, and 96 h (*n* = 3). Heatmap clustering showed that most untreated (DMSO) and toyocamycin biological repeats clustered according to their time of treatment, except for two out of three replicates of 2 h treated samples that clustered with control samples. Using volcano plot representations, we showed that 2 h toyocamycin treatment only led to 3.5% gene downregulation, whereas the level of differentially expressed genes reached up to 60% after longer toyocamycin exposure times from 10 h to 96 h (Appendix A). The number of read counts and the signal of expression levels (Log_2_FC) of differentially expressed genes increased over time, suggesting an important alteration in gene expression profiles (Figure 2B,C). As the duration of exposure to toyocamycin increased, the proportion of genes being upregulated also started to increase. After 96 h treatment, the number of genes that were upregulated following drug exposure became equal to those that were downregulated (Figure 2B,C). Interestingly, gene ontology analysis revealed that top downregulated pathways were associated with RNA polymerase II, suggesting that toyocamycin impacts a mechanism associated with the regulation of transcription (Figure 2D).

### 3.4. Cyclin-Dependent Kinase 9 Inhibition by Toyocamycin

To investigate the mechanistic insights of toyocamycin on gene regulation, we used connectivity mapping software (https://portals.broadinstitute.org/cmap/, accessed on 25^th^ May 2018) to associate the toyocamycin transcriptomic signature to gene expression profiles produced by known drugs [33]. We selected RNA-Seq data of YB5 cells treated cells at 250 nM toyocamycin for 10 h. This experimental condition was selected as a low cytotoxic treatment producing GFP reactivation and transcriptomic changes. The best hit, the transcription profile that correlated the most with the toyocamycin signature, was alsterpaullone, a known CDK1 inhibitor (Appendix A) [34]. Among the 10 top drugs, 4 CDK inhibitors were identified by connectivity mapping [35,36,37,38]. The other hits belonged to different classes of anticancer drugs such as inhibitors of protein synthesis, DNA topoisomerase, DNA methyltransferase, and DNA/RNA synthesis [35,39,40,41,42]. Based on the enrichment for CDK inhibition as a primary target, we hypothesized that toyocamycin could inhibit CDKs.

Based on our previous studies, we demonstrated that pharmacological inhibition or genetic downregulation of CDK1 or CDK2, which are the targets of some of the connectivity mapping hits, did not cause GFP reactivation in the YB5 system [15]. CDK1 and CDK2 belong to cell cycle CDK family members, which mainly impact cell cycle. Therefore, we focused on other family members, such as CDK7-13, which are known to mainly regulate gene transcription pause release machinery [43]. Previously, we characterized that specific CDK9 inhibition causes GFP reactivation in the YB5 system and discovered HH1, a specific CDK9 inhibitor [15]. To assess whether toyocamycin inhibits transcriptional CDKs, we compared RNA-seq data sets of toyocamycin and HH1 at different time points. We quantified the number of genes that were commonly differentially expressed in YB5 cells treated with either toyocamycin (250 nM) or HH1 (10 µM) for 2, 24, and 96 h (log2 fold change values less than −1 and more than +1, *p*-adjusted value < 0.05; Figure 3A–C). After 2 h, 293 genes were differentially expressed as compared to the untreated cells common to both drug treatments. Interestingly, almost all genes (99.3%) were significantly downregulated by both toyocamycin and HH1, suggesting rapid inhibition of gene transcription, a hallmark of CDK9 inhibition as a blocker of transcription elongation (Figure 3A). Only 0.7% of the differentially expressed genes showed opposite transcription profiles in response to each drug (upregulated by toyocamycin and downregulated by HH1), suggesting a similar transcriptional impact. After 24 h treatment, 2868 genes were differentially expressed as compared to untreated YB5 cells and were common to both treatments (Figure 3B). About 90% of the differentially expressed genes shared a similar profile after each drug with 55% of them downregulated by both treatments and 36% of them upregulated. Only a minority of differentially expressed genes (less than 10%) showed opposite gene expression behavior in response to toyocamycin or HH1, again supporting a similar transcriptomic profile of both drugs (Figure 3B). After 96 h treatment, 848 genes were differentially expressed as compared to untreated cells and were common in both treatments (Figure 3C). Again, the vast majority of the genes were similarly regulated (76.2%) by both drugs, i.e., 24.9% of them were downregulated and 51.3% were upregulated. Less than a quarter (24%) of differentially expressed genes showed opposite transcription regulation in response to toyocamycin or HH1. Interestingly, rapid transcription inhibition followed by a steady increase in gene expression in response to toyocamycin was also observed with HH1 because of CDK9 inhibition. Indeed, we associated this dual gene expression response to rapid CDK9 inhibition on RNA polymerase II elongation followed by chromatin opening and epigenetic reactivation due to the BRG1 loss of phosphorylation, as previously demonstrated [15]. Furthermore, CDK9 inhibition by HH1 triggered the reactivation of endogenous retroviruses (ERVs) in YB5 cells. We also observed that toyocamycin induced significant ERV expression after 24 h treatment to an extent similar to that of HH1 (97.6% similarity, 124 ERVs, Figure 3D). Therefore, we showed that the transcription signature triggered by toyocamycin resembled that of CDK inhibitors with a close similarity to HH1 [15].

### 3.5. Toyocamycin Modulates CDK9 Enzymatic Activity in Colon Cancer Cell Lines

To validate the impact of CDK9 and toyocamycin on GFP expression, we transfected *HA*- and *FLAG*-tagged CDK9 in YB5 cells (Figure 4A). Then, we measured by flow cytometry the impact of CDK9 overexpression on toyocamycin’s effect on *GFP* transgene reactivation (Figure 4B). Neither overexpressing vector (*HA-CDK9*, *FLAG-CDK9*) alone (DMSO-treated) triggered *GFP* expression. As controls, DNA hypomethylating drug (DAC, 50 nM, 72 h) or selective CDK9 inhibitor (MC180295, 500 nM, 48 h) induced GFP fluorescence in about 40% and 10% of YB5 cells, respectively [15]. We selected a low dose toyocamycin treatment (250 nM, 48 h) producing about 10% of GFP-expressing YB5 cells. CDK9 overexpression in conjunction with toyocamycin treatment resulted in a reduction in the number of GFP-expressing cells. Similar findings were observed with CDK9 inhibitor MC180295. Altogether, CDK9 overexpression competed with CDK9 inhibitor (MC180295) and toyocamycin at inducing GFP, suggesting an interaction between the drugs and the enzyme.

Then, we asked whether toyocamycin could modulate a well-characterized target of CDK9, i.e., RNA-Pol II Ser2-P. To investigate toyocamycin’s specificity on CDK9 inhibition vs. the cell cycle CDKs, we measured the impact of toyocamycin treatment on Rb phosphorylation level (T826), a classic target of cell cycle CDKs (CDK4/6). HCT116 colon cancer cells were exposed for 6 h to different doses of toyocamycin (50–500 nM) and to another CDK9 inhibitor (BAY1251152, 1–10 µM, 16 h) as control (Figure 4C). Levels of RNA Pol II, its phosphorylated form on serine-2, Rb, phospho-Rb, and CDK9 were determined by immunoblotting on whole cell protein extracts. Two different RNA Pol II antibodies (raised against N-terminal and C-terminal regions) were used. We noted that toyocamycin (above 350 nM) or BAY1251152 (1–10 µM) reduced the RNA Pol II band intensity when using the Pol II C-terminal antibody. The N-terminal antibody revealed two bands, a hyperphosphorylated band (top) and a hypophosphorylated band (bottom), showing that the hypophosphorylated band expression levels were not impacted by the treatment [44]. Interestingly, toyocamycin significantly reduced RNA-Pol II ser 2 phosphorylation at 350–500 nM, similar to the BAY1251152 (Figure 4C,D). Toyocamycin, at all doses, did not change CDK9 levels or Rb and its phosphorylated form, suggesting no effect on CDK9 protein levels or the CDK4/6 activity. To verify the impact of toyocamycin on transcriptional CDKs and not on the cell cycle CDKs, we measured cell viability after 24 and 72 h treatment from 50 nM to 50 µM in YB5 and HCT116 colon cancer cells (Figure 4E). Although toyocamycin reduced cell viability as compared to untreated cells, cell viability was still above 50% even at the highest doses, suggesting that toyocamycin does not trigger immediate cytotoxicity. However, long-term toxicity was highlighted by a full eradication of the clonogenic potential of cancer cells, two weeks after a 24 h treatment at 10 nM, suggesting a delayed effect on cancer cells (Appendix A). The impact of toyocamycin on cancer cell proliferation showed time- and dose-dependent effects, as measured in metastatic melanoma cells (451-Lu-BR cells) by trypan blue assays. Low doses of toyocamycin after 24 h treatment did not reduce cell proliferation, while the impact of toyocamycin on cancer cell proliferation increased after 48 h to a full proliferation arrest after 72 h treatment (Appendix A). As toyocamycin produced delayed cytotoxicity and cancer cell proliferation, we investigated whether toyocamycin would also have a delayed impact on cell cycle progression. We exposed HCT116 human colon cancer cells to 6 h toyocamycin treatment, which was sufficient to reduce CDK9 activity (as shown by a reduction in Phospho-Pol II level, Figure 4C). Then, toyocamycin was removed for cell culture media and cell culture was continued for an additional 18 h to allow cancer cell progression in the cell cycle, prior to cell cycle analysis. In these conditions, the cell cycle was not significantly impacted by toyocamycin as compared to DMSO-treated cells (Figure 4F). Only 10% of apoptotic cells was measured after treatment in these conditions. However, longer toyocamycin treatment (24 h) significantly blocked the cell cycle (reduction in G0/G1, increase in G2/M), indicating that toyocamycin has a delayed effect on the cell cycle (Appendix A). Thus, the anticancer effects of toyocamycin do not appear immediately as observed with CDK4/6 inhibitors [45], suggesting that toyocamycin anticancer effects may be mediated in part by the downstream transcriptomic effects resulting from CDK9 inhibition. Overall, toyocamycin shows potent and selective CDK9 inhibitory activity and delayed anticancer effects.

### 3.6. Toyocamycin Shows CDK9 Inhibition Selectivity in Kinase Assays

To expand our studies on toyocamycin-induced CDK inhibition, we used in vitro kinase assays and performed dose-response curves. We obtained IC_50_ values generated with several CDKs bound to their cyclin ligands (Figure 5). Firstly, against our main target CDK9/CycT1, toyocamycin showed an IC_50_ value of 79.5 nM (Figure 5A). Then, we tested the effect of toyocamycin on CDK7/Cyclin H/MAT1, another transcriptional CDK, demonstrating 35 times less enzymatic inhibition (IC_50_ = 2.8 µM, Figure 5B) as compared to CDK9. Lastly, we investigated the inhibitory effects of toyocamycin against three cell cycle CDK complexes, CDK2/cyclin 2A, CDK4/cyclin D3, and CDK6/cyclin D3, which showed IC_50_ values of 0.67 µM, 15 µM, and > 10 µM, respectively (Figure 5C,D). Interestingly, the activity of known CDK2 or CDK4/6 small molecule inhibitors more potently inhibited the cell cycle CDK than toyocamycin (Figure 5C,D). Collectively, toyocamycin produced selective enzymatic CDK9 inhibition as compared to CDK7, but more importantly to cell cycle CDKs, suggesting a more pronounced effect on transcription elongation than the cell cycle.

### 3.7. Toyocamycin Adopts a Specific Position in the CDK9 Catalytic Site Compared to Other CDKs

To evaluate the toyocamycin binding pose within the different CDKs, we performed unbiased (without a reference position) molecular docking simulations, using the CDK4 model structure as well as the crystal structures of CDK2, CDK6, CDK7m and CDK9, bound to small molecule inhibitors from PDB. The rationale of using protein structures bound to a ligand was to ensure that the CDKs were in a conformation more likely to accommodate a ligand [46]. We compared the best docking pose, i.e., the toyocamycin pose, with the best CF value on each CDK (Figure 6A). The position of toyocamycin in CDK9 shows a CF value of −256.4 AU, suggesting a broadly favorable interaction between the ligand and the protein. Toyocamycin adopts almost exactly the same binding position on CDK7, for which the best docking pose has a CF value of −209.2 AU (Figure 6A). For CDK4 and CDK6, a similar binding mode on the adenosine core of toyocamycin is observed, while their sugar moiety adopts very different conformations. The binding to CDK4 and CDK6 shows good affinity with CF values of −152.9 AU and −235 AU, respectively. As for CDK2, toyocamycin binds in a unique conformation but still with a good binding affinity (CF = −248.9 AU). Altogether, the data suggest that toyocamycin binds to CDK ATP binding pockets but with a specific position and potentially stronger binding affinity on transcriptional CDKs (CDK9–7) as compared to cell cycle CDKs (CDK2, 4, 6).

CDKs have been crystalized in many forms with many small molecule inhibitors, except with toyocamycin. However, one crystal structure with toyocamycin is available in the PDB in complex with Rio1 protein kinase (PDB ID 3RE4) [47]. Rio1 kinase is a member of the Rio kinase family, involved in ribosome biogenesis and playing a role in 40S ribosomal subunit maturation, where toyocamycin binds directly into its ATP binding pocket [47]. As expected, there is a low protein sequence similarity between Rio1 and any of the CDKs tested in the kinase assays (CDK2, −4, −6, −7, and −9). However, the tri-dimensional (3D) superimposition of Rio1 and each of the CDKs demonstrated a 3D structural similarity with overall RMSD values ranging from 4.4 Å to 4.8 Å (Appendix A). Since the ATP binding sites of Rio1 and each CDK have a similar 3D spatial arrangement, we determined the binding mode of toyocamycin using Rio1 binding as a reference position. First, we transferred the toyocamycin crystal pose in Rio1 into each CDK and calculated an associated CF value, called reference CF (REF_CF, Appendix A). REF_CF value calculations showed a positive value for all the CDKs, except for CDK9 and CDK4 (Appendix A). These findings demonstrate that the crystal position of toyocamycin in Rio1 does not have a favorable position in CDK2, 6, and 7 because of steric clashes with binding site residues, but could be in a more favorable position in CDK9 and 4.

Then, we performed molecular docking simulations of toyocamycin starting from the Rio1 crystal pose as a reference and compared the best docking solution with the reference pose on each CDK (Figure 6B, Appendix A). We evaluated the closest possible binding pose to the Rio1 binding mode (best RMSD) on each of the CDKs and associated CF values. CDK9 and CDK7 showed RMSD values of 1.5 Å and 1.1 Å, respectively, suggesting that the docking predicted a binding pose of toyocamycin in the CDKs close to the Rio1 pose. This approach (using Rio1 binding mode as reference) gave a lower CF score to CDK7–9 compared to the docking without any reference (as shown in Figure 6A). Interestingly, CDK4 and 6 did not exhibit favorable binding, illustrated by CF values of −64 AU and −40.9 AU, respectively (Figure 6B, Appendix A). As for CDK2, docking simulations showed that toyocamycin could not adopt a Rio1 binding mode where its closest pose is in a completely different conformation with a best RMSD value of 2.3 Å and CF of −156 AU (Figure 6B, Appendix A). The binding mode of toyocamycin on CDK9 and CDK7 as defined in Figure 6A is in agreement with the experimentally determined pose of toyocamycin in Rio1 kinase [47]. Therefore, using two different docking approaches, we validated the position of toyocamycin into several CDK catalytic pockets but with a similar binding pattern in CDK9 and CDK7 (Figure 6A,B).

Leveraging on the computational predictions of the binding mode of toyocamycin, we then determined the putative interacting residues of toyocamycin within the ATP binding site of CDK9 (Figure 6C). The amino acids which contribute the most to the binding mode are Ile25, Phe30, Val33, Ala46, Phe103, Asp104, Phe105, Cys106, Ala153, Leu156, Ala166, and Asp167. As a comparison, we also show the interacting residues determined experimentally between toyocamycin and the Rio1 crystal structure (Figure 6C). Although toyocamycin exhibits similar binding poses in CDK9 and CDK7 (Figure 6A,B), it showed 35 times more enzymatic inhibition against CDK9 as compared to CDK7 (Figure 5A,B), suggesting a different binding affinity. Thus, we determined the binding of toyocamycin in the CDK7 catalytic site. As expected, the toyocamycin binding mode in CDK7 involved a different series of amino acids including Leu18, Val26, Ala39, Phe91, Asp 92, Phe93, Met94, Asn141, Leu144, Asp155, and Asp212 (Figure 6D). While CDK7 and CDK9 exhibit sequence similarities in their catalytic sites (Appendix A), the 3D structure of the ATP binding site is different between the two isoforms, resulting in a different binding mode of toyocamycin. Therefore, the difference between the bound amino acids and the spatial arrangement of the binding site could potentially explain the lower enzymatic inhibition of toyocamycin in CDK7 as compared to CDK9. In summary, docking simulations confirmed the preferential binding of toyocamycin in the CDK9 catalytic site with a unique pose involving a specific series of amino acids.

The selectivity of toyocamycin for CDK9 within the CDK protein family members raises the question of how small differences in the sequence contribute to different drug sensitivity responses. Indeed, pharmacological selectivity at the binding site may be influenced by small sequence differences in the protein backbone, which trigger a 3D conformation change directly into the catalytic pocket. To explore the impact of the protein backbone on the catalytic site as a determinant of toyocamycin activity, we used the cases of CDK9, CDK2, and CDK6, which showed high differences in toyocamycin inhibitory effects in kinase assays with IC_50_ values of 0.08 µM, 0.67 µM, and > 10µM, respectively (Figure 5A,C,E). We mutated in silico all amino acids in the ATP binding site to develop chimeric model structures of CDK9 to mimic CDK2 (denoted as CDK9MCDK2) and CDK6 (denoted as CDK9MCDK6) exactly followed by docking simulations (Appendix A). We also achieved reversed mutations where CDK2 and CDK6 have modified ATP binding site mimicking the one of CDK9, referred to as CDK2MCDK9 and CDK6MCDK9, respectively. Then, we performed a docking simulation and retained the best CF values of each mutant (Appendix A). Similar docking poses and CF values between mutated and wild-type CDKs imply a weak contribution of the backbone protein to the overall binding of toyocamycin, whereas a different prediction supports that widespread small alterations of the backbone impact toyocamycin drug efficacy. First, the toyocamycin pose in CDK9MCDK2 shows a minor difference only for the ribose ring (salmon color), whereas the rest of the molecule was in the same position as in the wild-type CDK9 (green color, CF = −256; RMSD = 0.2 Å; Appendix A). Similarly, the toyocamycin pose was almost identical between CDK9 (green) and CDK9MCDK6 (CD = −265; RMSD = 0.6Å; Appendix A). Together, these results suggest that the backbone conformation of the binding site of CDK9 is favorable for toyocamycin binding. Reversed mutations using CDK2MCDK9 and CDK6MCDK9 (adding CDK9 catalytic site amino acids into the other CDK catalytic sites) demonstrated a different binding pose for CDK2MCDK9 (CF = −156; RMSD = 0.7 Å) and a similar pose in CDK6MCDK9 (CF = −213; RMSD = 0.4 Å) (Appendix A), suggesting the importance of both catalytic site residues and protein backbone contribution to the overall binding of toyocamycin. Overall, the in silico mutations and docking simulations support the importance of not only the catalytic site but also the protein backbone binding site to the 3D structural arrangement and the consequent toyocamycin affinity. Altogether, our docking simulations confirmed that toyocamycin is a preferential inhibitor of CDK9 binding in the ATP catalytic pocket. While the favorable binding is due to the unique arrangement of its binding pocket, this comes as a result of not only the specific amino acids present in its binding site, but also as a result of slight backbone rearrangements due to variations in the sequence of CDK9 compared to other kinases.

## 4. Discussion

Through drug screening and target identification studies, we report that toyocamycin, a natural compound previously known for its potent anticancer effects and with reticulum endoplasmic stress activity, showed selective CDK9 inhibitory effects in the low nanomolar range [22,31,48,49]. As compared to approved epigenetic drugs (DNA methylation and HDAC inhibitors), toyocamycin produced comparable gene reactivation and anticancer effects but through a different epigenetic mechanism. To delineate its pharmacological target, we explored its transcriptomic effects, which showed a rapid onset of transcription inhibition followed by a time-dependent increase in the number of downregulated and upregulated genes. We previously observed this drug-induced transcriptional pattern after the treatment with a selective CDK9 inhibitor [15]. Gene set enrichment analysis, and connectivity mapping also pointed towards a mechanism involving CDK inhibition. Then, we compared the toyocamycin transcriptional signature at different time points to the one of HH1, a novel and specific CDK9 inhibitor which we recently discovered [15]. Interestingly, the results showed a close correlation between transcriptional profiles. In addition, we showed that toyocamycin produced a significant decrease in RNA-Pol II serine 2 phosphorylation, a canonical target of CDK9 enzymatic activity [43,50,51,52]. Interestingly, toyocamycin treatment did not influence Rb phosphorylation levels; a common target of cell cycle CDKs, suggesting that toyocamycin is selective to CDK9. Enzymatic kinase assays confirmed that toyocamycin preferentially inhibited CDK9 with an IC_50_ value of 79.5 nM, which was the lowest among other transcriptional and cell cycle CDKs. Lastly, we used docking simulations to provide an insight on toyocamycin selectivity towards CDK9 as compared to other CDKs, which have strong protein sequence similarities. Our analyses revealed that toyocamycin is bound into the ATP catalytic site of CDKs but harbors a unique pose in CDK9 as a consequence of the specific nature of amino acids throughout the sequence of CDK9 within and outside of the binding site, which may explain its potency and selectivity towards CDK9. Sangivamycin, a toyocamycin analogue which produced weak GFP activation in our model (Appendix A), was reported to mediate direct CDK9 inhibition, further supporting our findings [53].

CDKs are a family of serine-threonine kinases which form complexes with various cyclin subunits to control cell proliferation and cell death. CDK isoforms are separated into cell cycle CDKs (CDK1, −2, −4, and −6), transcriptional CDKs (CDK7, −8, −9, −12, −13, and −19), and other regulators with functions that remain to be better characterized (CDK5, −10, −11, −14, −15, −16, −17, −18, and −20). Here, we focused on CDK9, which was discovered by Graña et al. [54]. CDK9 is a nuclear protein and forms a heterodimer with regulatory cyclins T1, T2a, or T2b. This complex represents the main component of the positive transcription elongation factor b (P-TEFb). It mediates gene transcription elongation through phosphorylation of serine 2 residues of the Y_1_S_2_P_3_T_4_S_5_P_6_S_7_ heptapeptide repeats in the C-terminus domain (CTD) of RNA polymerase II [50,51]. Thus, CDK9 is a critical kinase for productive transcription. High CDK9 expression level correlates with poor prognosis and drug resistance in several cancer types including with breast cancer, lung cancer, prostate cancer, endometrial cancer, melanoma, osteosarcoma, myeloid leukemia, and soft tissue sarcoma [51].

CDK9 inhibition produces a drastic reduction in global mRNA production and also downregulates highly expressed genes that are essential for cancer cell proliferation, metastasis, and treatment resistance. These genes, including the oncogenic transcription factors *MYC* and *MYCN* or the apoptosis regulator *MCL-1*, are commonly downregulated after CDK9 inhibition [55,56,57]. In addition to its role of triggering transcription elongation, CDK9 enzymatic activity is also required for the maintenance of heterochromatin compaction by a mechanism involving CDK9-mediated phosphorylation of BRG1 and the silencing of tumor suppressor genes [15]. Altogether, CDK9 plays key roles in controlling gene expression through transcription elongation regulation and maintaining epigenetic silencing, representing an interesting target for cancer chemotherapy [15,43,52].

Targeting CDK9 in cancer showed promising results in preclinical and clinical research [56]. The first CDK9 inhibitor tested in a clinical trial was flavopiridol (alvociclib) based on its preclinical anticancer activities and effective CDK9 inhibition, yet it exhibited cross activity with cell cycle CDKs [51]. Indeed, due to the similarities in the catalytic sites amongst all CDKs, small-molecule selectivity is an issue, which led to unsuccessful clinical trials [56]. A flavopiridol derivative, namely dinaciclib, was generated with improved CDK9 specificity (as compared to other CDKs) and is currently being tested in clinical trials. A new generation of CDK9 inhibitors with higher specificity and lower toxicities, allowing their use in monotherapy or in combination therapy, are actively being developed. Most of them, including CAN-508, LDC000067, SNS-032, and AZD-4573, produce CDK9 inhibition with IC_50_ values in the low nanomolar range, causing apoptosis and downregulation of MCL-1 in leukemia and in tumor cell lines (breast, osteosarcoma, ovarian cancer, prostate, melanoma, and endometrial cancers) [51]. Some of these compounds were tested in clinical trials [51,58,59]. Unfortunately, many of these inhibitors did not demonstrate significant anticancer effects in patients and were discontinued because of severe side effects. However, the 3rd generation of CDK9 inhibitors (BAY-1251152 and AT-7519) produced promising outcomes in patients with hematological malignancies and solid tumors, which will be evaluated further in future clinical trials [51,60].

Toyocamycin was tested in a phase I clinical trials (NSC-63701) in the mid-sixties based on promising anticancer effects against cancer cell lines without a clear understanding of its mechanism of action [61]. Twenty-three cancer patients received toyocamycin at 10–200 µg/kg for 5 days (by intravenous infusion over 1–2 h). No systemic toxic effects were observed, whereas local toxicity at the site of injection (severe phlebitis) was observed in patients receiving the highest doses.

## 5. Conclusions

Our study defines toyocamycin as a potent and selective CDK9 inhibitor at low doses. This natural product could be used as a small molecule tool to modulate CDK9 activity in vitro and its specific binding could spark some interest to design novel CDK9 inhibitors.

## Figures and Tables

**Figure 1 cancers-14-03340-f001:**
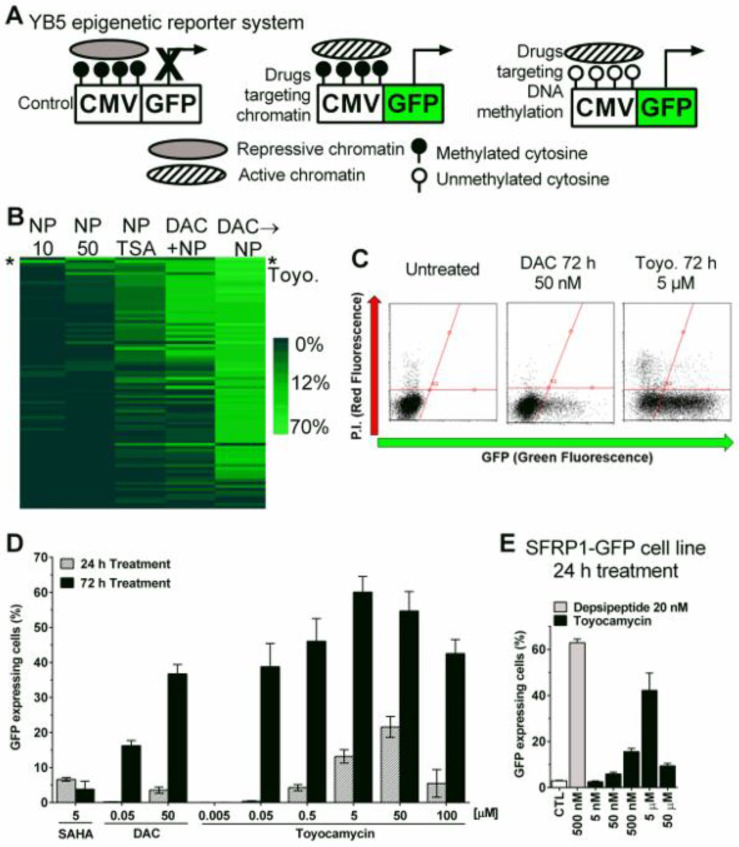
Epigenetic drug screening with natural compound library identifies toyocamycin as the most promising hit. (**A**) Schematic description of YB5 epigenetic reporter system in untreated condition (left), showing GFP expression upon drug-induced active chromatin (middle), or after drug-induced DNA demethylation and consequent chromatin activation (right). (**B**) Heatmap showing percentages of GFP-expressing cells after natural products (NP) screening in YB5 cells with the 5 different schedules (*n* = 1). NP10 means treatment with natural products at 10 µM for 72 h. NP50 means treatment with natural compound library at 50 µM for 24 h. NP TSA means treatment with natural products at 10 µM for 72 h followed by 24 h treatment with 200 nM trichostatin A. DAC+NP means 72 h treatment with decitabine (DAC) in simultaneous combination with 10 µM NP. DAC → NP means 72 h treatment with decitabine at 50 nM followed by NP at 50 µM. Each horizontal line represents a compound. Toyocamycin is highlighted with a star on the heatmap (*). (**C**) Flow cytometry graphical representations of YB5 cells for GFP fluorescence (x-axis) counterstained with propidium iodide (P.I., *y*-axis) for dead cell staining in untreated and treated cells (drugs and doses are indicated on the graphs, 10,000 cells were acquired). (**D**) Percentage of GFP-expressing cells in YB5 cells after vorinostat (SAHA), DAC, and toyocamycin treatments (at doses indicated on the graph, *n* = 3). (**E**) Percentage of GFP-expressing cells in HCT116 cells after depsipeptide and toyocamycin treatments (at doses indicated on the graph, *n* = 3).

**Figure 2 cancers-14-03340-f002:**
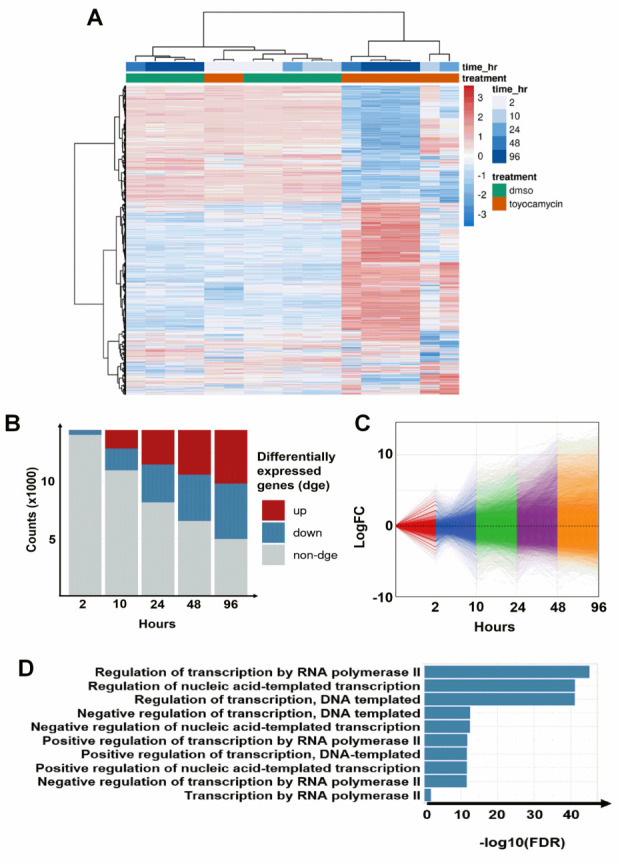
Toyocamycin induces potent time-dependent gene expression changes. YB5 cells were treated with toyocamycin (250 nM) during 2, 10, 24, 48, and 96 h prior to RNA-seq. (**A**) Heatmap showing gene expression log2 fold change (blue-red scale) between toyocamycin-treated and untreated groups (16,535 genes are shown on the heatmap). The time points at 2 and 10 h are done in duplicates. The time points at 96 h have been done in triplicates and the 24/48 h of treatment have been done with only one replicate. (**B**) Number of differentially expressed genes (upregulated in red and downregulated in blue, unchanged expression is depicted in grey). After 2 h toyocamycin treatment, 513 genes were downregulated and 24 genes were upregulated. After 10 h toyocamycin treatment, 2299 genes were downregulated and 1999 genes were upregulated. After 24 h toyocamycin treatment, 3651 genes were downregulated and 3387 genes were upregulated. After 48 h toyocamycin treatment, 4377 genes were downregulated and 4449 genes were upregulated. After 96 h toyocamycin treatment, 5061 genes were downregulated and 5050 genes were upregulated. (**C**) Gene expression variation over time during toyocamycin treatment. (**D**) Gene enrichment analysis showing RNA regulatory pathways that are downregulated after toyocamycin treatment (2 h).

**Figure 3 cancers-14-03340-f003:**
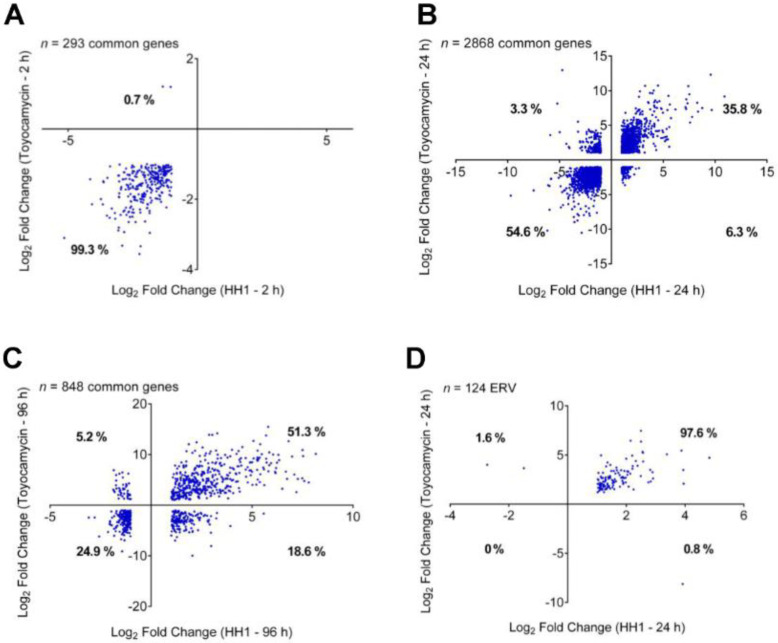
Common gene regulation between toyocamycin treatment and HH1 treatment in YB5 cells. RNA-seq data sets were merged to compare the expression values of common genes differentially regulated by toyocamycin and HH1 in YB5 cells (downregulated genes with Log2FC < –1 and upregulated genes with Log2FC > 1, *p*-adjusted values < 0.05). Gene expression data sets are presented (**A**) 2 h, (**B**) 24 h, and (**C**) 96 h after toyocamycin exposure. (**D**) Endogenous retrovirus expression levels were compared between toyocamycin and HH1 treatments (24 h). The number of common genes is indicated on all the graphs.

**Figure 4 cancers-14-03340-f004:**
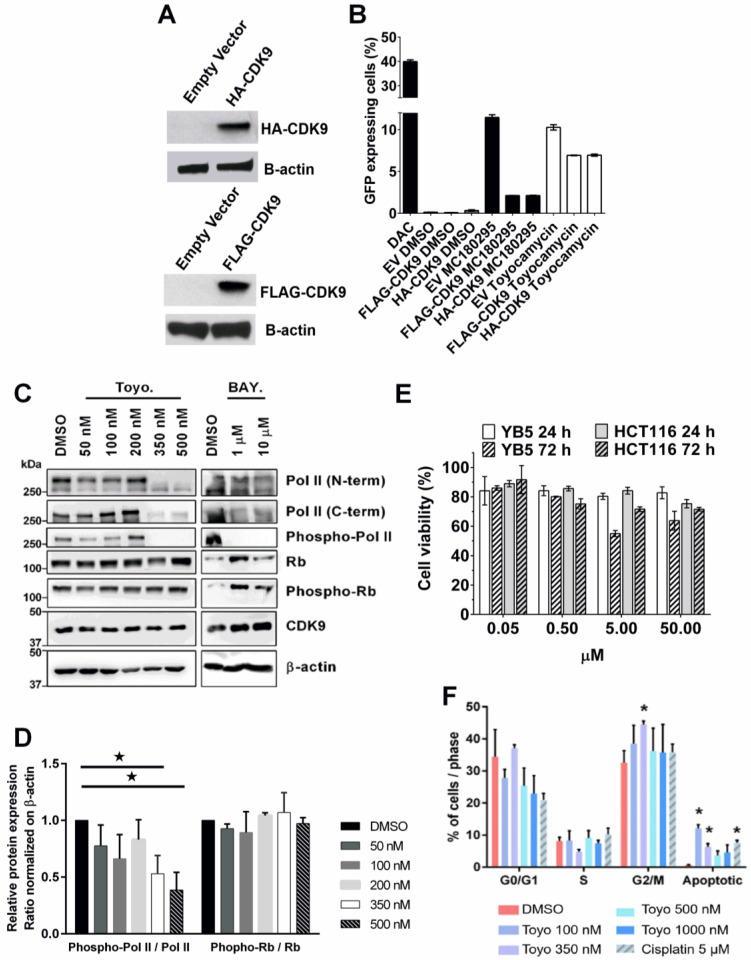
Toyocamycin demonstrates CDK9 inhibition in colon cancer cells. YB5 cells were transfected with HA-CDK9 or FLAG-CDK9 vectors. (**A**) Western blotting shows transfection efficiency. Loading control is shown using β-actin antibody. (**B**) Percentage of GFP-expressing YB5 cells after DNA hypomethylating drug (DAC, 50 nM, 72 h) or selective CDK9 inhibitor (MC180295, 500 nM, 48 h) in presence or absence of CDK9-expressing vectors (*n* = 2). (**C**) Protein expression levels of CDK9 and cell cycle CDK targets. RNA-Pol II (N-term and C-term), phospho-Pol II, Rb, phospho-Rb, and CDK9 levels were measured after 6 h toyocamycin treatment in HCT116 cells (doses are indicated in the graph) or with DMSO. Treatments with CDK9 inhibitor BAY 1251152 at 1–10 µM for 16 h were used as a positive control. β-actin was used as loading control. Molecular weight of each protein is indicated on the graph. Full western blots are available in supplementary. (**D**) Ratio of Phospho-Pol II Ser 2/Pol II and Phospho-Rb T826/Rb, normalized on β-actin and DMSO are shown (mean  ±  SEM, N ≥ 3 biological replicates, *: *p* ≤ 0.05 obtained by unpaired Student’s *t* test). (**E**) Cell viability analysis by Viacount staining in YB5 and HCT116 cells after 24 and 72 h toyocamycin treatment (0.05, 0.5, 5, and 50 µM, *n* = 3) relative to untreated cells. (**F**). Cell cycle analysis by flow cytometry in HCT116 cells after 6 h toyocamycin treatment followed by 18 h without exposure (*n* = 3, a star indicates statistical difference between treated group and control, unpaired Student’s *t*-test, *p* < 0.05).

**Figure 5 cancers-14-03340-f005:**
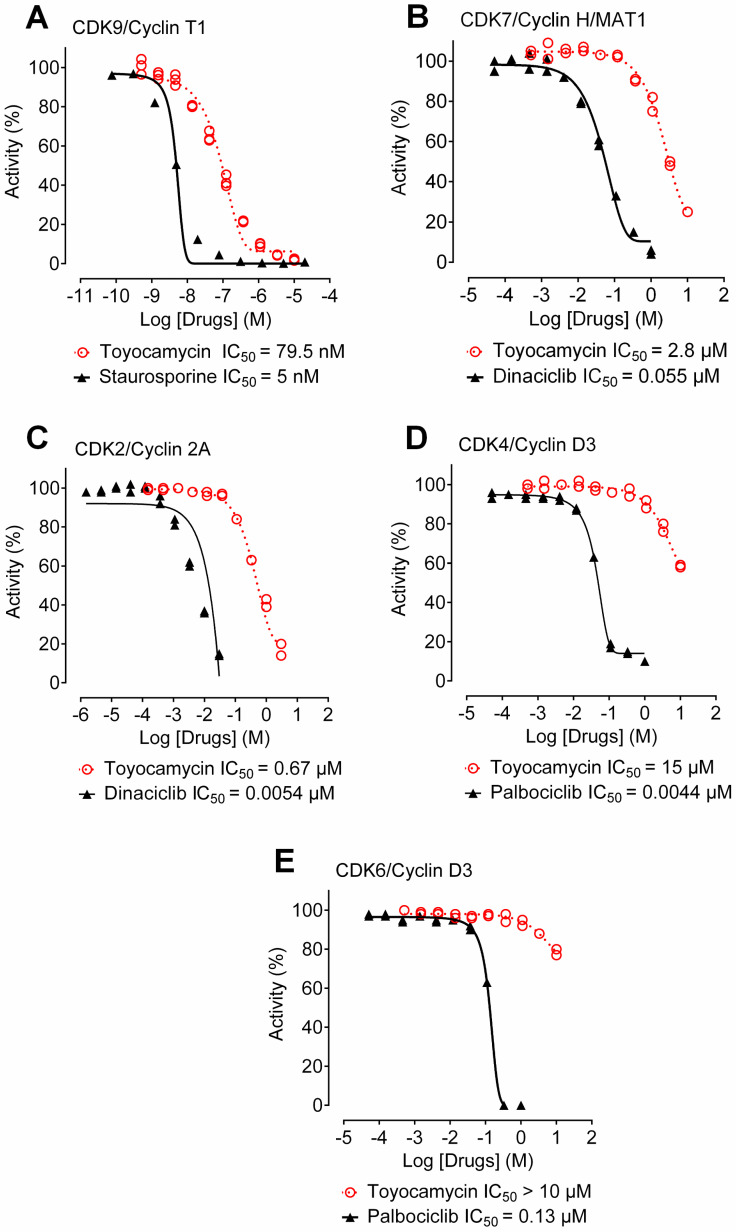
Toyocamycin is a potent and specific inhibitor of CDK9. Enzymatic assays were performed with toyocamycin at different concentrations against (**A**) CDK9/CyclinT1, (**B**) CDK7/Cyclin H/MAT1, (**C**) CDK2/Cyclin 2A, (**D**) CDK4/Cyclin D3, and (**E**) CDK6/Cyclin D3. Concentration producing 50% enzymatic inhibition (IC_50_) was calculated for toyocamycin and other inhibitors listed in the graphs.

**Figure 6 cancers-14-03340-f006:**
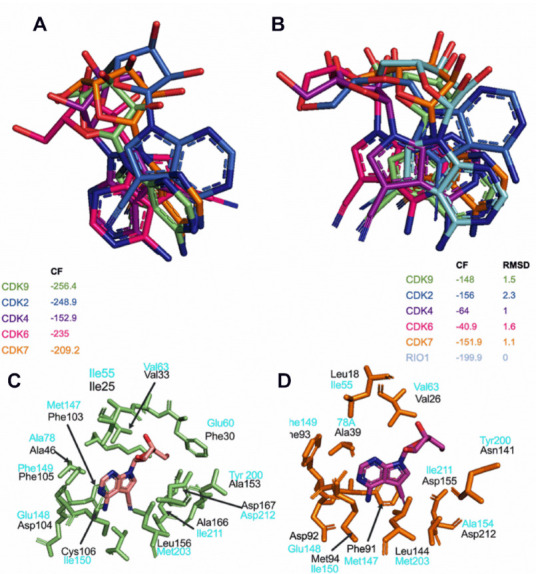
Toyocamycin docking simulations in different CDKs. (**A**) Best docking poses of toyocamycin in CDK2 (blue marine), CDK4 (purple), CDK6 (pink), CDK7 (orange), and CDK9 (lime green). (**B**) Best RMSD poses of toyocamycin within Rio1 kinase crystal pose in the different CDKs. (**C**) ATP binding site residues interacting with toyocamycin in CDK9 (lime green) and (**D**) CDK7 (orange) with corresponding interacting residues determined in Rio1-toyocamycin complex (light blue).

## Data Availability

The datasets generated during and/or analyzed during the current study are available from the corresponding author upon reasonable request. The RNA-seq data generated in this study are available in GEO under the accession number GSE202744.

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
