# Peer review of "Selective CDK9 Inhibition by Natural Compound Toyocamycin in Cancer Cells"

_cancers, 2022, doi:10.3390/cancers14143340_

Round 1
Reviewer 1 Report
The manuscript entitled “Selective CDK9 inhibition by natural compound toyocamycin in cancer cells” reports that toyocamycin can be used as a small molecule tool to modulate CDK9 activity and its specific binding can design novel CDK9 inhibitors. This finding provides valuable information for the development of promising anticancer agents and therapies. However, the following points must be addressed before the manuscript can be suitable for publication.
Comments:
- In methods, the authors need to describe the version of softwares used (ImageJ, FlexAID, EdgeR..).
- The authors need to indicate the total number of genes analyzed and the number of high or low expressed genes in Figure 2A.
- The authors should indicate the size of the protein markers in Figure 4C.
- The authors should show the results of cell cycle analysis.
- Although this study performed the docking simulation analysis, the authors need to confirm whether toyocamycin directly binds to CDK9 through several binding experiments such as IP, DARTS, CETSA, or SPR, to clarify that toyocamycin is a selective inhibitor of CDK9.
- To propose toyocamycin as an epigenetic anticancer drug, the anticancer activity of toyocamycin should be assessed using an in vivo animal tumor model.
- In discussion, the authors indicated that toyocamycin was 79.4 nM, the lowest among CDKs, but in Figure 5A it is indicated as 79.5 nM, which should be corrected (lines 636-637).
Author Response
Comments from Reviewer 1
1. Describe the version of softwares used (ImageJ, FlexAID, EdgeR..).
Response: We have provided a description of version of the softwares used in the manuscript in the Materials and Methods section.
2. The authors need to indicate the total number of genes analyzed and the number of high or low expressed genes in Figure 2A.
Response: We indicated in the Figure 2A legend (p.26) the total number of genes analyzed in the heatmap (16,535 genes). The information is added to the figure 2A legends. In addition, we gave more detailed information regarding the number of down and upregulated genes in the legend of Figure 2B.
3. The authors should indicate the size of the protein markers in Figure 4C.
Response: We have indicated the size of the protein markers in Figure 4C.
4. The authors should show the results of cell cycle analysis.
Response: We provide cell cycle analysis on colon cancer cells (HCT116) in Figure 4F showing that toyocamycin does not alter cell cycle for short term treatments (6h treatment followed by 18h without treatment prior to cell cycle analysis). A significant increase in apoptotic cells was measured (sub-G1) even at low doses (up to 10%). As expected, a longer exposure time (24h) of toyocamycin (500nM or 1µM) induced a cell cycle block in G0/G1 (Supplementary Fig.S6B). Cell cycle analysis indicates that toyocamycin does not block immediately cell cycle suggesting a weak impact on cell cycle CDKs.
5. Although this study performed the docking simulation analysis, the authors need to confirm whether toyocamycin directly binds to CDK9 through several binding experiments such as IP, DARTS, CETSA, or SPR, to clarify that toyocamycin is a selective inhibitor of CDK9.
Response: We thank the reviewer for this interesting suggestion. Although these binding experiments may clarify toyocamycin by a different methodology, these experiments will take months prior to set up and optimize. Indeed, binding identification methods exhibit biases in proteome coverage (Theranostics 2022; 12(4):1829-1854), which is associated with false discovery. However, we believe that the integration of four independent experimental approaches from in vitro kinase CDKs inhibition (Fig.5), western blotting of CDKs targets (Fig.4C-D), connectivity mapping (Table S1) and docking analyses (Fig.6; FigS7-8, Table S2-S3) provide sound evidences of direct binding of toyocamycin to CDK9. Further supporting our findings, sangivamycin, a toyocamycin analogue, which showed GFP activation in our YB5 cell system (Supplementary Fig S2) was reported to mediate direct CDK9 inhibition (Dolloff NG, Allen JE, Dicker DT, Aqui N, Vogl D, Malysz J, et al. Sangivamycin-like molecule 6 exhibits potent anti-multiple myeloma activity through inhibition of cyclin-dependent kinase-9. Molecular cancer therapeutics 2012;11(11):2321-30 doi 10.1158/1535-7163.MCT-12-0578).
6. To propose toyocamycin as an epigenetic anticancer drug, the anticancer activity of toyocamycin should be assessed using an in vivo animal tumor model.
Response: As an alternative to in vivo animal model, we now show in Figure 4 and Supplementary FigS6A, the impact of toyocamycin on cancer cell viability in 2 colon cancer cell lines and one metastatic melanoma cell line.
7. In discussion, the authors indicated that toyocamycin was 79.4 nM, the lowest among CDKs, but in Figure 5A it is indicated as 79.5 nM, which should be corrected (lines 636-637).
Response: We have corrected the typo on p19.
Reviewer 2 Report
In this manuscript, the authors have performed drug screen with their GFP cell-based system in natural product library (120 compounds) and identified toyocamycin as a potential epigenetic drug. Furthermore, they explored the possible mechanism of toyocamycin and identified it as a specific CDK9 inhibitor. Next, they evaluated the binding mode of toyocamycin and CDK9 with molecular docking simulation. Overall, they suggested that toyocamycin could be used to module CDK9 activity and spark interest for the development of novel CDK9 inhibitors. However, there are some remaining questions to be answered:
1, some typo errors, including
- Line 128, inconsistent font size;
- Line 475, “phosphor-Rb”;
- Line 477, “um”;
- Line 556, extra comma after “Figure 6C,”
- Line 669, extra space after “flavopiridol”.
2, The experimental conditions described in Materials and Methods and Results are not matched in scheme shown in Fig S1A.
3, Could the authors describe how many replicates they performed for some assays?
4, In Fig S1, No matter which schedule was used, Toyocamycin treatment (10uM or 50uM) only caused less than 15% GFP positive cells. However, the authors claimed that more than 60% GFP positive cells after toyocamycin treatment (Fig 1C-D). Could the authors explain the discrepancy?
5, The resolution of Fig S5 is too low to tell the number shown on the volcano plots.
6, The authors indicated that toyocamycin interacted with CDK9, not other cell cycle CDK family members. Have the authors examined the cell cycle distribution with toyocamycin treatment?
7, Based on previous studies, toyocamycin is more than just a CDK9 inhibitor, it is possible that toyocamycin exhibited anti-tumor effect via other mechanism. Could the authors discuss this possibility?
Author Response
Comments from Reviewer 2
1. Correction of some typo errors, including on line 128, inconsistent font size; on line 475, “phosphor-Rb”; on line 477, “um”; on line 556, extra comma after “Figure 6C,” on line 669, extra space after “flavopiridol”.
Response: All the typos are corrected.
2. The experimental conditions described in Materials and Methods and Results are not matched in scheme shown in Fig S1A.
Response: We corrected the order to the different experimental conditions in the text to match the one in the Fig S1A on p.6.
3. Could the authors describe how many replicates they performed for some assays?
Response: We included the number of repeats where it was missing in the figure legends or in the method sections. Briefly, the drug screen was performed one time (Fig1.D). For drug validations, experiments were performed 3 times (Fig1.C-D). For RNA-seq, one or up to three samples per condition were harvested and sequenced. Experiments of the FLAG-/HA-tagged CDK9 over-expression were done in duplicate (Fig4A). The western blotting on Fig 4C-D were repeated 3 times. New figures 4E-F, cell viability and cell cycle data were repeated 3 times. The enzymatic assays in Figure 5 were repeated three times for CDK9 and two times for the rest of the CDKs (information added in the method section).
4. In Fig S1, No matter which schedule was used, Toyocamycin treatment (10uM or 50uM) only caused less than 15% GFP positive cells. However, the authors claimed that more than 60% GFP positive cells after toyocamycin treatment (Fig 1C-D). Could the authors explain the discrepancy?
Response: The results obtained with during the initial screen were performed using a drug library. However, all other experiments were performed using pure toyocamycin compound, which show more potent activity during in vitro validation. Thus, compound purity could explain a stronger activity as compared to the effect of compounds from the drug library.
5. The resolution of Fig S5 is too low to tell the number shown on the volcano plots.
Response: We thank the reviewer for the comment. We have increased the size of the numbers on the volcano plots and on the axes.
6. The authors indicated that toyocamycin interacted with CDK9, not other cell cycle CDK family members. Have the authors examined the cell cycle distribution with toyocamycin treatment?
Response: We showed in the enzymatic assays (Fig.5) that toyocamycin shows a stronger inhibition on CDK9 (IC50 79 nM) as compared to other CDK (IC50 between 600 nM to more than 10 µM). To address the reviewer’s question, we performed cell cycle analysis on HCT116 cells treated with toyocamycin for 6h (6h treatment followed by 18h without drug allowing the cells complete cell cycle, Fig4F) and 24h (Suppl Fig S6B) at different doses. As expected with short time exposure (6h), toyocamycin weakly impacted cell cycle (Fig 4F). However, longer exposure time (24h) impacted cell cycle, which could be the results of the inhibition of other cell cycle CDK or the downstream transcriptomic effects resulting from CDK9 inhibition (Suppl FigS6B).
7. Based on previous studies, toyocamycin is more than just a CDK9 inhibitor, it is possible that toyocamycin exhibited anti-tumor effect via other mechanism. Could the authors discuss this possibility?
Response: Yes, we agree with this statement. We had included a reference referring to toyocamycin effect on XP1 pathway stress responses in multiple myeloma (reference 22).
Reviewer 3 Report
Dear authors,
This study is good and needs to be improved for our audience. It contains more lack of literature evidence which is not possible to complete the story here and validated methodology and need to be improving further to build this study strong enough to publish.
Although, this manuscript needs to be improved through the rewrite improvement and typo errors all in the manuscript. Please take care of all the manuscript text again and include some recent references and cancer related publications to support your study in Selective CDK9 inhibition by natural compound toyocamycin in cancer cells, which is still missing.
- why the authors did not mention how many cells are per wells they used as per their text in the line “YB5 cells were seeded at 30,000 cell/ml in 96 well-plates and treated 3 days”. During the 3 days how they measure the quantity of cells and their treatment strategy with their target molecule which is not clear.
- In figure 1C is not clear and does not look good. Provide best and high resolution to the difference.
- Figure 4 A is not similar band for b actin in western blot. Why?
Author Response
Comments from Reviewer 3
1. Please take care of all the manuscript text again and include some recent references and cancer related publications to support your study in Selective CDK9 inhibition by natural compound toyocamycin in cancer cells, which is still missing.
Response: As requested by the reviewer, we have corrected typos and included 2 references supporting the potential of CDK9 inhibition in cancer research (Diamond JR, Boni V, Lim E, Nowakowski G, Cordoba R, Morillo D, et al. First-in-Human Dose-Escalation Study of Cyclin-Dependent Kinase 9 Inhibitor VIP152 in Patients with Advanced Malignancies Shows Early Signs of Clinical Efficacy. Clin Cancer Res 2022;28(7):1285-93 doi 10.1158/1078-0432.CCR-21-3617.) and the direct inhibition of CDK9 mediated by sangivamycin, an analogue of toyocamycin (Dolloff NG, Allen JE, Dicker DT, Aqui N, Vogl D, Malysz J, et al. Sangivamycin-like molecule 6 exhibits potent anti-multiple myeloma activity through inhibition of cyclin-dependent kinase-9. Molecular cancer therapeutics 2012;11(11):2321-30 doi 10.1158/1535-7163.MCT-12-0578).
2. Why the authors did not mention how many cells are per wells they used as per their text in the line “YB5 cells were seeded at 30,000 cell/ml in 96 well-plates and treated 3 days”. During the 3 days how they measure the quantity of cells and their treatment strategy with their target molecule which is not clear.
Response: We indicated the number of cells seeded per well (6,000 cells/well) in the method section on p 6. To clarify the treatment strategy for cell viability, proliferation assays and cell cycle analysis, we added a paragraph in the method section (page 8).
3. In figure 1C is not clear and does not look good.
Response: We have removed the fluorescence images and enlarged the flow cytometry plots and redone the axes of the plots for better resolution.
4. Figure 4 A is not similar band for b actin in western blot. Why?
Response: The experiments (FLAG- and HA-tagged CDK9 over expression) were run on two different gels with different protein extracts, which explains why the bands are not exactly identical.
Round 2
Reviewer 1 Report
The uploaded manuscript is not a final revised version. So, it is difficult to easily check the revised figures and contents of this paper. Please send it back with the final revision.
Reviewer 3 Report
Dear authors,
Overall work here, I believe the authors have done an exemplary job in preparing this manuscript.